# Use of machine learning and principal component analysis to retrieve nitrogen dioxide (NO$_2$) with hyperspectral imagers and reduce noise in spectral fitting

Joanna Joiner[1], Sergey Marchenko[2], Zachary Fasnacht[2], Lok Lamsal[3], Can Li[4], Alexander Vasilkov[2], and Nickolay Krotkov[1]

[1]National Aeronautics and Space Administration, Goddard Space Flight Center, Laboratory for Atmospheric Chemistry and Dynamics, Code 614, Greenbelt, MD USA
[2]Science Systems and Applications, Inc., Lanham, MD USA
[3]University of Maryland, Baltimore County, MD USA
[4]University of Maryland, College Park, MD USA

**Correspondence:** Joanna Joiner(joanna.joiner@nasa.gov)

**Abstract.** Nitrogen dioxide (NO$_2$) is an important trace-gas pollutant and climate agent whose presence also leads to spectral interference in ocean color retrievals. NO$_2$ column densities have been retrieved with satellite UV-Vis spectrometers such as the Ozone Monitoring Instrument (OMI) and Tropospheric Monitoring Instrument (TROPOMI) that typically have spectral resolutions of the order of 0.5 nm or better and spatial footprints as small as 3.6 km $\times$ 5.6 km. These NO$_2$ observations are used to estimate emissions, monitor pollution trends, and study effects on human health. Here, we investigate whether it is possible to retrieve NO$_2$ amounts with lower spectral resolution hyper-spectral imagers such as the Ocean Color Instrument (OCI) that will fly on the Plankton, Aerosol, Cloud, ocean Ecosystem (PACE) satellite set for launch in early 2024. OCI will have a spectral resolution of 5 nm and a spatial resolution of $\sim$1 km with global coverage in 1-2 days. At this spectral resolution, small scale spectral structure from NO$_2$ absorption is still present. We use real spectra from the OMI to simulate OCI spectra that are in turn used to estimate NO$_2$ slant column densities (SCDs) with an artificial neural network (NN) trained on target OMI retrievals. While we obtain good results with no noise added to the OCI simulated spectra, we find that the expected instrumental noise substantially degrades the OCI NO$_2$ retrievals. Nevertheless, the NO$_2$ information from OCI may be of value for ocean color retrievals. OCI retrievals can also be temporally averaged over time-scales of the order of months to reduce noise and provide higher spatial resolution maps that may be useful for downscaling lower spatial resolution data provided by instruments such as OMI and TROPOMI; this downscaling could potentially enable higher resolution emissions estimates and be useful for other applications. In addition, we show that NNs that use as inputs coefficients of leading modes of a principal component analysis of radiance spectra appear to enable noise reduction in NO$_2$ retrievals. Once trained, NNs can also substantially speed up NO$_2$ spectral fitting algorithms as applied to OMI, TROPOMI, and similar instruments that are flying or will soon fly in geostationary orbit.

 # 1 Introduction

Nitrogen dioxide ($NO_2$) is an important trace gas for both air quality and climate. It is identified as a criteria pollutant by the United States (US) Environmental Protection Agency (EPA). As a climate agent, it is a precursor for tropospheric ozone, a potent greenhouse gas in the upper troposphere. $NO_2$ also contributes in the formation of aerosols that can cool the planet by reflecting incoming solar radiation back to space (Shindell et al., 2009). Over non-polluted regions, most of the atmospheric column of $NO_2$ resides in the stratosphere, where it participates in photochemical reactions that can affect the ozone layer (see e.g., van Geffen et al., 2020, and references therein).

Much effort has been expended to develop sophisticated physically-based retrieval algorithms for spectrometers that measure scattered solar radiation at ultraviolet (UV) and blue wavelengths at the ground (e.g., Noxon, 1975; Platt and Perner, 1983; Platt, 1994; Platt and Stutz, 2006) as well as from satellite platforms (e.g., Burrows et al., 1999; Richter and Burrows, 2002; Bucsela et al., 2006; Boersma et al., 2007, 2011; Valks et al., 2011; Bucsela et al., 2013; Yang et al., 2014; Marchenko et al., 2015; Boersma et al., 2018; van Geffen et al., 2020; Lamsal et al., 2021). Retrievals from satellite-based instruments such as the Global Ozone Monitoring Experiment (GOME) (Burrows et al., 1999), SCIAMACHY (Bovensmann et al., 1999), the Ozone Monitoring Experiment (OMI) (Levelt et al., 2006), GOME-2 (Munro et al., 2016), the Ozone Mapping and Profiler Suite - Nadir Mapper (OMPS-NM) (Bak et al., 2017), and the TROPOspheric Monitoring Experiment (TROPOMI) (Veefkind et al., 2012) have been used in numerous studies related to top-down emissions estimates, air quality monitoring and forecasting, pollution events, trends, and related health studies (see e.g., Bovensmann et al., 2011; Lamsal et al., 2015; Krotkov et al., 2016; Duncan et al., 2016; Levelt et al., 2018; Goldberg et al., 2021; Kerr et al., 2021; Cooper et al., 2022, and references therein).

$NO_2$ absorption impacts satellite radiance measurements that are used in ocean color algorithms. The affected spectral ranges include those used for retrievals of colored dissolved organic matter (CDOM) and chlorophyll-a (e.g., Mannino et al., 2008; Le and Hu, 2013). In particular, large variations in $NO_2$ total columns near polluted coastlines can affect ocean color measurements from sensors in both low Earth orbit (LEO) and Geostationary Earth Orbit (GEO) (Ahmad et al., 2007; Tzortziou et al., 2014). For example, under high $NO_2$ loading ($\sim 1 \times 10^{16}$ molec cm$^{-2}$), if not accounted for, errors in water leaving radiance could reach 10-20% (Ahmad et al., 2007) and produce spectral structure that can interfere with ocean color retrievals.

Two planned hyper-spectral imagers, the Ocean Color Instrument (OCI) on the Plankton, Aerosol, Cloud, ocean Ecosystem (PACE) mission in LEO and the GEO Geosynchronous Littoral Imaging and Monitoring Radiometer (GLIMR) were designed for ocean color measurements. The PACE OCI is scheduled to launch in the early 2024 time frame (Werdell et al., 2019), and GLIMR will make diurnal measurements over the Gulf of Mexico and surrounding coastlines later this decade (NASA, 2019). There are several options for atmospheric $NO_2$ correction algorithms for ocean property retrievals: 1) use of a satellite-based $NO_2$ climatology; 2) use of collocated satellite data from atmospheric instruments, such as TROPOMI; 3) use of simulated $NO_2$ data from a chemistry transport model. However, all of these have shortcomings. For example, $NO_2$ derived from atmospheric instruments may not be available at the spatial or temporal scales of ocean color instruments. While a climatology or model simulations can capture the basic features of $NO_2$ including high values around coastal cities, they may miss important details of pollution plumes that can extend over the ocean. It should also be noted that while TROPOMI and PACE will be in similar

orbits (TROPOMI and PACE equator crossing times are 13:30 and 13:00, respectively), TROPOMI will be nearing its end of primary mission by the time PACE launches; its follow-on mission, Sentinel 5, will be in a morning orbit (09:30 equator crossing time) that will result in more temporal mismatch with PACE. Therefore, it has been a stated and challenging goal of these ocean color missions to quantify $NO_2$ spatio-temporal variations with their lower spectral resolution measurements. $NO_2$ retrievals have been demonstrated with hyperspectral sensors on aircraft (Tack et al., 2017; Kuhlmann et al., 2022) and the Russian Resurs-P satellite (Postylyakov et al., 2017, 2019; Zakharova et al., 2021); these sensors have somewhat higher spectral resolution than PACE and GLIMR.

This paper attempts to answer two questions: 1) Can $NO_2$ slant columns be accurately estimated with planned ocean sensors such as PACE OCI and GLIMR using a machine learning algorithm?; 2) Can the machine learning techniques developed for retrieving $NO_2$ from the ocean color instruments be harnessed to improve existing algorithms applied to atmospheric instruments, both in terms of quality and efficiency? If the answer to question 1 is affirmative and retrievals from PACE OCI and GLIMR are successful, they could provide improved spatial resolution of $NO_2$ total column amounts as compared with existing instruments, a benefit for land and ocean retrievals as well as for atmospheric science. Regarding question 2, state-of-the-art spectral fitting algorithms can be computationally burdensome. Algorithm efficiency is particularly important for current and future atmospheric composition instruments, including those in geostationary orbit, that have very large data volumes. Machine learning has been shown to be an efficient means of estimating $NO_2$ vertical columns from satellite spectra (Li et al., 2022) as well as for other applications in remote sensing and geoscience (Maxwell et al., 2018; Lary et al., 2016). In addition, machine learning combined with principal component analysis may be able to reduce noise in the spectral fitting as compared with the more traditional approaches.

## 2 Data and methods

### 2.1 OMI, PACE, and GLIMR satellite instrument characteristics

Table 1 gives a summary of the approximate relevant characteristics of the satellite instruments used in this study and other relevant sensors used for $NO_2$ retrievals. OMI is a push broom spectrometer that measures backscattered sunlight and solar irradiance (Levelt et al., 2006, 2018). There are three separate detectors on OMI. We ran experiments with L1B Earthshine radiances from the OMI collection 3 for the visible (VIS) (Dobber, 2007b) and UV-2 detectors (Dobber, 2007a) that cover wavelengths from 349–504 nm and 307–383 nm, respectively. OMI employs a two dimensional (2D) charged coupled device (CCD) that provides spectral information in one dimension and spatial information in the other. This results in sixty rows of spectra in the cross track direction. Spacecraft motion provides observations along the satellite swath. Therefore, each cross track row of OMI can be considered as a distinct instrument with its own characteristics (wavelengths, response functions, calibration) and biases. The wavelengths vary slightly across the swath resulting in a so-called spectral smile. We use data from early in the mission (2005) for this study. Later in the mission, some of the rows were affected by an anomaly presumably outside the instrument that caused blockage and scattering of light into some of OMI's rows (Levelt et al., 2018) resulting in a decrease of spatial coverage. OMI's spatial resolution is approximately 13 km in the along track direction by 24 km in the

**Table 1.**

| Satellite/Instrument | spec. range (nm) | spec. resolution (nm) | spec. sampling (nm) | spat. resolution (nadir, km) | coverage |
|---|---|---|---|---|---|
| [a] Aura OMI Vis detector | 349–504 | 0.63 | 0.21 | 13×24 | global, 1–2 days |
| [b] Sentinel 5P TROPOMI Vis detector | 400–496 | 0.54 | 0.20 | 3.6×5.6 | global, 1 day |
| [c] PACE OCI | 340-890 | 5 | 2.5 | 1×1 | global, 1–2+ days |
| [d] GLIMR | 340-1040 | 10 | 5 | 0.3×0.3 | regional coastlines, sub-daily |

[a] Schenkeveld et al. (2017)

[b] van Geffen et al. (2022)

[c] Werdell et al. (2019)

[d] Mannino, private communication (2022)

cross track direction at nadir with larger pixels towards the swath edges. The total swath width is $\sim$2600 km. The TROPOMI instrument has similar spectral characteristics for $NO_2$ retrievals but with higher spatial resolution. PACE OCI will cover wavelengths from UV ($\sim$340 nm) through the short-wave infrared wavelengths. It will provide daily global coverage from LEO at a spatial resolution of approximately 1 km (Werdell et al., 2019). GLIMR will have a spatial resolution of about 300 m.

Figure 1a shows the specified signal-to-noise ratio (SNR) for GLIMR provided by the instrument team (A. Mannino, priv. comm., 2022); we assume these values are independent of radiance value. Figure 1b shows an SNR model for PACE OCI based on measurements, where the SNR varies with radiance, also provided by the instrument team (B. Franz et al., priv. comm., 2022). Typical radiance distributions for different wavelengths are shown in Appendix A. Note that for determination of $NO_2$ tropospheric vertical columns, cloudy observations are used to help estimate stratospheric column amounts, so that the full range of radiance values is needed, not just clear-sky observations (Bucsela et al., 2013).

## 2.2 $NO_2$ absorption cross sections and DOAS retrievals

$NO_2$ absorption covers a broad range of wavelengths from the UV to the near-infrared (NIR) with a peak near 400 nm. Figure 2 shows $NO_2$ absorption cross sections from the UV through red wavelengths (Vandaele et al., 1998), where the blue and green curves have been convolved and resampled to approximate effective cross sections for OMI and OMPS, respectively. The red and black curves show the $NO_2$ cross sections convolved with boxcar functions of widths 5 and 10 nm and 2 samples per box, similar to the expected spectral resolution and sampling of OCI and GLIMR, respectively. Particularly at the OCI spectral resolution, there is still marked high frequency structure throughout the visible wavelength range.

Most $NO_2$ spectral fitting algorithms are based on the differential optical absorption spectroscopy (DOAS) methodology (e.g., Platt and Stutz, 2006). In a DOAS-type spectral fit, the retrieved quantity is the slant column density (SCD) of a weakly absorbing trace gas, defined as the integrated number of molecules per unit area to produce an observed amount of absorption

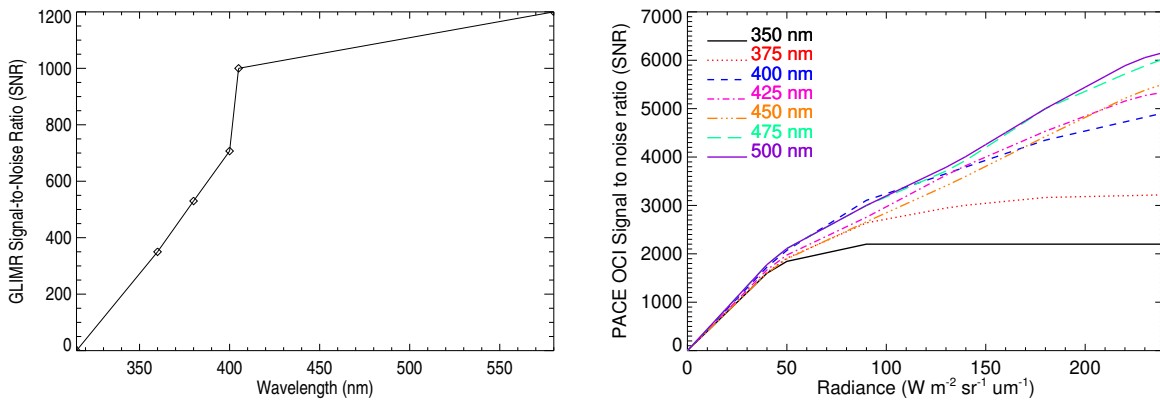

**Figure 1.** a) GLIMR specified SNR assumed constant with radiance; b) PACE OCI SNR as a function of wavelength and radiance based on measurements provided by the instrument team.

at a particular wavelength along the total atmospheric photon path. DOAS generally works by fitting appropriately convolved absorption cross-sections (effective cross-sections) to the logarithm of sun-normalized radiance spectra for a given fitting window. A DOAS slant column retrieval for $NO_2$ involves fitting the high frequency structure in the $NO_2$ absorption cross
sections generally in the range 400–497 nm (see Figure 2) while accounting for the low frequency envelope of $NO_2$ absorption, e.g., using a polynomial function (e.g., Richter and Burrows, 2002; Lerot et al., 2010; Richter et al., 2011; Marchenko et al., 2015; van Geffen et al., 2015, 2020). At GLIMR spectral resolution, a retrieval would likely need to make use of the broad $NO_2$ absorption feature peaking at around 400 nm rather than the finer spectral features used in DOAS retrievals. While this type of approach has been achieved for ozone (Fleig et al., 1986) whose stratospheric column amount is typically quite large,
it has not be demonstrated for gases with weaker absorption (owing in part to lower column amounts) such as $NO_2$.

Within any type of spectral fit, other known absorbers or pseudo-absorbers, such as rotational-Raman scattering (also known as the Ring effect), should be accounted for. In typical $NO_2$ fitting windows, the interfering absorbers include ozone ($O_3$), water vapor ($H_2O$), the oxygen dimer ($O_2-O_2$), and glyoxal ($CHO-CHO$). The spectral signature of the Earth's surface must also be accounted for along with other instrumental effects such as spectral alignment of the radiance spectra and proper
characterization of the instrument response function.

A secondary retrieval step involves estimation of the vertical column density (VCD) of $NO_2$ using the SCD, sun-satellite geometry, and information about clouds, aerosols, the Earth's surface, and the $NO_2$ profile shape. The SCD and VCD are related through the concept of an air mass factor (AMF), i.e., VCD = SCD / AMF. Both the SCD and AMF are formally wavelength dependent, so that typical DOAS retrievals using a range of wavelengths to fit a single value SCD or compute an
125 AMF refer to an average over the fitting window. Richter et al. (2014) have accounted for this wavelength dependence, but this type of approach is typically not used in operational algorithms. A direct VCD spectral fitting algorithm was also applied to UV wavelengths to retrieve $NO_2$ from the OMPS-NM (Yang et al., 2014).

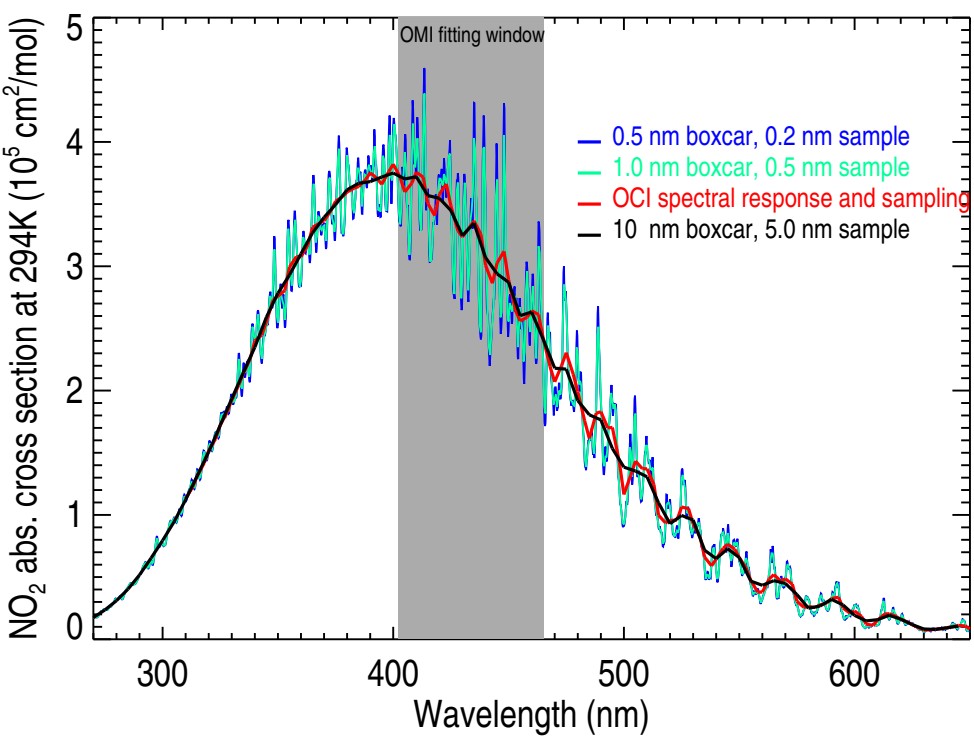

**Figure 2.** NO$_2$ cross sections from Vandaele et al. (1998) convolved with boxcar functions of different widths: Blue: similar spectral characteristics to OMI (though the OMI range limited to wavelengths <500 nm; green: similar to OMPS-NM; red: similar to OCI; black: similar to GLIMR.

## 2.3 Data flow and retrieval

Figure 3 shows a flow diagram of the data processing that we use here to train and evaluate results from a neural network
(NN) that predicts NO$_2$ SCDs using simulated data from imagers, such as OCI and GLIMR, with lower spectral resolution as
compared with spectrometers, such as OMI and TROPOMI, designed for atmospheric measurements. We simulate OCI and
GLIMR observations by reducing noise, then spectrally averaging and resampling OMI radiances and adding noise according
to instrument specifications and measurements (see Fig. 1). A machine learning approach is then employed to estimate the
OMI-derived NO$_2$ slant columns with simulated data from the lower spectral resolution instruments. As explained in more
detail below, we focus on NO$_2$ SCD retrievals. The conversion of SCD to total or tropospheric VCD can be accomplished in a
straight-forward and computationally efficient manner as in current algorithms and is not addressed further here. Information
about the surface, aerosol, and clouds, such as cloud/aerosol radiance fraction and effective pressure (e.g., from the O$_2$ A band),

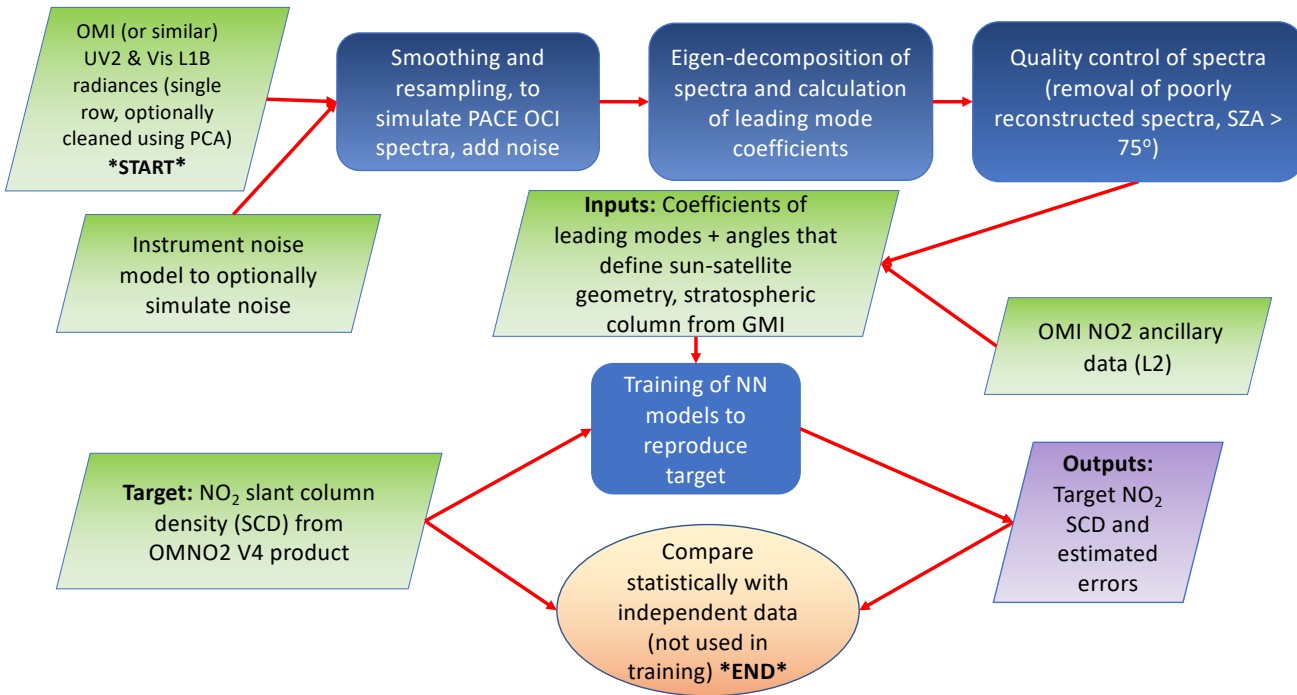

**Figure 3.** Data flow diagram showing simulation of radiance for a low spectral resolution hyper-spectral instrument (PACE OCI) using observations from a higher spectral resolution instrument (OMI) as well as training and evaluation of a neural network (NN) to estimate $NO_2$ slant column densities (SCDs).

needed for the calculation of the AMF, will be available from OCI itself (Werdell et al., 2019). Details of the input features, target outputs, and architecture of the machine learning algorithm are provided in the following subsections.

### 140    2.3.1   OMI $NO_2$ data

We use the destriped SCDs (parameter name SlantColumnAmountNO2Destriped) from the collection 3 version 4.0 OMNO2 $NO_2$ product (Lamsal et al., 2021; Krotkov et al., 2019) as the NN training target. The $NO_2$ spectral fitting algorithm is based on Marchenko et al. (2015) who use an iterative approach in which the 402–465 nm range is broken up into seven smaller overlapping micro-windows. This method leads to flexible determination of wavelength-dependent shifts between radiance 145 and irradiance spectra as well as the rotational Raman scattering or so-called Ring spectrum. The overall OMI standard $NO_2$ product has undergone substantial changes over the years after being evaluated in numerous studies with respect to other model-, ground-, air-, and satellite-based data sets (e.g., Lamsal et al., 2014; Choi et al., 2020, and references therein).

### 2.3.2 Simulated OCI and GLIMR data

We start with the OMI Level 1B (L1B) radiances. Unlike in standard OMI retrieval algorithms, we do not perform any normalization with respect to either an observed or reconstructed (laboratory) solar reference spectrum. While this normalization could be done and there may be advantages, such as for processing long time series in which instrument degradation occurs, here we elected to keep the approach as simple as possible.

The next step is to optionally reduce noise in the OMI data using a principal component analysis (PCA) approach, where spectra are reconstructed using coefficients of leading principal components (PCs) constructed from a large sample of data. Here as in standard DOAS fits, we use the natural logarithm of the radiances, but without normalization as discussed above. The goal of the noise reduction is to provide a relatively clean, though not necessarily perfect, set of spectra to simulate data for different instrument configurations. Our aim is to produce a realistic simulation of satellite observations. While we could have used simulated spectra instead of OMI data, it would be difficult to capture all of the complexities present in real satellite-observed spectra, including instrumental artifacts, and other complex interactions involving scattering and absorption in the atmosphere and surface.

The data sample used in the initial PCA starts with all observations from one day in every month (the 15th day) in 2005 supplemented with additional days in winter when the $NO_2$ lifetime is long and pollution can build up in the boundary layer to give very high SCDs. The added high pollution days are 29 January through 04 February of 2005. The same data sample is used for training and evaluating the NN models as described below. Use of a large sample spanning different seasons assures that we cover a wide range of sun-satellite geometries as well as pollution, cloud, and surface conditions in the training data.

We found by trial and error that 60 PCs sufficiently reconstructs the OMI spectra in the range 349-503 nm (742 spectral samples) for the purposes of $NO_2$ spectral fitting, while removing spurious features that can occur for example within the south Atlantic anomaly region as described by Gorkavyi et al. (2021). In other words, beyond this number, there is not significant correlation between PCs and $NO_2$ absorption cross sections. The cleaned OMI L1B data are then averaged over different spectral bandpasses to simulate data from lower spectral resolution instruments. Here, we used a simple boxcar function with widths 5 and 10 nm and resampled at 2.5 and 5 nm to simulate OCI-, and GLIMR-like instruments, respectively. This produces 59 samples for the OCI-like instrument for a spectral range 355–502.5 nm. Finally, noise following a Gaussian distribution is optionally added to the simulated spectra at the same wavelength grid for all rows according to instrument specifications and measurements. Here, we assume that errors are not correlated with wavelength as information on correlated errors was not provided. Correlated errors could possibly degrade the performance of the retrieval if the neural network is not able to effectively account for them.

### 2.3.3 Machine Learning Architecture

The machine learning algorithm was constructed within the Interactive Data Language (IDL) software package. It consists of a three layer feed-forward artificial NN with two hidden layers and 1.3N nodes in each layer, where N is the number of inputs (Schmidhuber, 2015). Details regarding the inputs are given below. The output layer has OMI $NO_2$ SCD as a single node. We

use a soft-sign activation function for the first layer, a logistic (sigmoid) for the second layer, and a bent identity for the third layer. An adaptive moment estimation optimizer minimizes the error function with a learning rate of 0.1. We scale all inputs and outputs such that means are zero and standard deviations are equal to unity.

### 2.3.4 NN inputs

We perform a PCA (or eigen-decomposition) of the simulated spectra using the same data sample as described above for the noise reduction. The coefficients of a leading number of PCs are used as inputs to the NN. We find that the NN training converges faster when coefficients of the PCs are used rather than the measured radiances themselves, even when coefficients of all modes are used as inputs. The PCA concentrates on the spectral features corresponding to information about the atmosphere and surface in the leading modes while projecting the random instrument noise onto the trailing modes. This may make it easier

for the NN to reject those coefficients with little information content pertaining to the target. We found, by trial and error, that maintaining a number of PCs equal to half the number of spectral elements was sufficient to capture the spectral information associated with $NO_2$ while providing some noise reduction. Since the trailing PCs typically express random spectral noise, eliminating these modes can lead to noise reduction.

 We then perform quality control on the spectra. We remove all data with slant columns less than zero or greater than

$1 \times 10^{17}$ molec cm$^{-2}$. We also remove any pixels where the solar zenith angle (SZA) is greater than 75°. Finally, we check that the quality flag on the OMI SCD data indicates a good fit.

 The NN training is performed separately for each OMI CCD detector row, because each row has unique spectral characteristics. While it is possible to perform NN training on all rows at once, since the data are spectrally averaged to a uniform wavelength grid as in Fasnacht et al. (2022), we find that slightly better performance is achieved by training on each row

individually.

 The inputs to the NN are then the coefficients of the leading PCs and other optional parameters that may aid the NN in trying to match the target OMI $NO_2$ SCD output variable. An important consideration for selection of input parameters is that we are training a NN to estimate SCDs produced by a DOAS-like algorithm that used a more narrow fitting window weighted towards the blue spectral region as noted in Figure 2. SCDs, because they depend upon the atmospheric photon path, are by definition wavelength dependent. UV wavelengths have less sensitivity to lower tropospheric $NO_2$ than blue wavelengths owing to the

effects of Rayleigh scattering that increase towards the UV and generally reduce the amount of light reflected from the surface (Richter et al., 2014).

 We tested several parameters that can help to determine how much of the OMI-derived SCD originates from the lower troposphere where UV wavelengths have less sensitivity. These include the cosine of the solar and view zenith angles, cosine

of the scattering angle, the stratospheric $NO_2$ column from the Global Modeling Initiative (GMI) chemical transport model, the geometry-dependent Lambertian-equivalent reflectivity (GLER) of the surface (Vasilkov et al., 2017, 2018; Qin et al., 2019; Fasnacht et al., 2019), surface pressure, and the effective scene pressure that is related to both the cloud pressure and optical thickness. Over ocean, the GLER accounts for the anisotropy of solar reflection from the ocean surface and backscatter from the bulk of ocean water and over land accounts for the bidirectionality of scattered sunlight from shadowing in vegetation. We also

tried to use the stratospheric column provided by the Global Modeling Initiative (GMI) model multiplied by the stratospheric air mass factor, or in other words the expected stratospheric SCD based on the model estimate.

The parameters that were ultimately selected as input features are shown in Fig. 3. We find that the combination of stratospheric column $NO_2$ and the cosines of solar zenith and scattering angles slightly improve the estimates of the target OMI $NO_2$ SCDs as discussed in more detail below. The cosine of the view zenith angle is nearly constant for a given row and does 220  not provide significant improvement. The other inputs tested similarly did not substantially improve the fitting. The spectra themselves contain information about these variables, although the training of the NN may require more iterations if these variables are not included as predictors. For example, information about the cloud optical thickness and underlying surface is present in the radiances and can be disentangled using machine learning (Joiner et al., 2022; Fasnacht et al., 2022). Information about cloud pressure is contained within the oxygen dimer absorption band near 477 nm (Acarreta et al., 2004) as well as from 225  the filling-in of solar Fraunhofer lines by rotational-Raman scattering (e.g., Joiner and Bhartia, 1995). We take a minimalist approach here with respect to the predictors and below only include stratospheric column amount and the cosines of solar zenith and scattering angles in addition to coefficients of leading PCs as features in the results shown below.

### 2.3.5  NN outputs

We also tested a variety of different target outputs. We tested both $NO_2$ SCD and the natural logarithm of the $NO_2$ SCD as 230  target outputs. We find that slightly better results are obtained using the natural logarithm of the $NO_2$ SCD, likely because SCDs are more normally distributed in log space, which is desirable for neural network training. BoxCox transformations (Box and Cox, 1964) may produce slightly better results yet, but will depend upon the sample used. As the distributions of $NO_2$ have undergone changes with time, particularly in polluted regions, results obtained by training on a single distribution may not be optimal for a given time period.

We also tried training directly on the total VCD. More iterations for training may be needed owing to complex relationships with atmospheric constituents and the Earth's surface that impact the photon pathlength and that are needed to estimate the VCD including dependencies on clouds, aerosol, and the surface bi-directionality. Without detailed knowledge of the $NO_2$ profile (we used only estimates of the stratospheric and tropospheric columns from the GMI), we did not obtain a satisfactory result and for simplicity focus on SCD exclusively as the target parameter below. The conversion from SCD to VCD can be 240  achieved efficiently using existing algorithms.

### 2.3.6  NN training and evaluation

For the training, we used every third data point from the sample described above (i.e., using data from at least one day in every month), providing more than 100,000 samples for each row. We then compare statistical results including variance explained ($r^2$), bias defined as the mean of $SCD_{\text{true}}$ - $SCD_{\text{est}}$, where $SCD_{\text{true}}$ and $SCD_{\text{est}}$ are the true (from OMNO2) 245  and estimated $NO_2$ SCDs, respectively, and root-mean-squared difference (RMSD = $\sqrt{\left(\sum_{n=1}^{N}\left[SCD_{n,\text{true}} - SCD_{n,\text{est}}\right]^2\right)/N}$) based on independent samples (i.e., not used in the training set). We did not find evidence of overtraining. We also use a completely independent day (28 January 2005) for visual evaluation below.

## 2.4 Addressing instrumental noise in the training process

There are several possible ways of dealing with the effects of instrumental noise in the training and application of NNs. One method is to train a NN using noiseless data, then apply the trained network to noisy data. Another method is to train a NN using noisy data. The latter approach is likely to provide the best result as the NN learns how to properly weight the different wavelengths based on a large sample that provides information about the wavelength dependence of the SNR. The former approach may work well if the SNR is relatively constant with wavelength. Another advantage to the former approach is that only one trained network is needed for either simulations or application to real data where the inputs could have a variety of different SNRs.

To mitigate the impact of instrumental random noise, one may either spatially average the noisy SCD retrievals or spectrally average radiance observations from adjacent fields-of-view (FoV) and use the coefficients computed with averaged spectra as inputs to the trained NN. If there are spatially dependent biases in the SCD results obtained with noisy data, these biases will not be eliminated by averaging together noisy retrievals. Therefore, it may be more advantageous to average spectra together and present the averaged data as inputs to a NN. The disadvantage to this approach, as described above, is that for optimal results, one may need to train a separate NN for each SNR scenario. We tested all of these approaches and found that the best results were obtained by averaging spectra together and training separate NNs for each SNR scenario. In practice, spatially averaging of pixels would be employed to increase the SNR, thus degrading the spatial resolution of the resulting retrievals. This is the approach taken by e.g. Tack et al. (2017) with the Airborne Prism EXperiment (APEX) hyper-spectral sensor, where spectra were averaged over an array of $20{\times}20$ pixels (a ground sample distance of $60{\times}80\,\mathrm{m}^2$) to increase the SNR to 2500. This SNR value gave SCD uncertainties of $3.4$–$4.4{\times}10^{15}\,\mathrm{molec\,cm^{-2}}$ with their limited fitting window of 470–510 nm and FWHM values of 2.4–3.4 nm at the center wavelength of 490 nm. This relatively small fitting window was used to estimate $NO_2$ SCDs owing to interference from instrumental artifacts and/or other atmospheric spectral contaminants at other wavelengths where there is strong $NO_2$ absorption. Postylyakov et al. (2017) similarly averaged spectra from a hyper-spectral sensor on the Resurs-P/2 satellite to a spatial resolution of 2.4 km to provide precision of about $2.5{\times}10^{15}\,\mathrm{molec\,cm^{-2}}$ for SCD. They used a fitting window covering 420-490 nm for their retrieval with a spectral resolution that varies from 2.5–4 nm over this range.

## 3 Results

### 3.1 Results with simulated PACE OCI spectra

We tested approaches initially using spectra from OMI UV-2 and Vis channels with a range of 325–502.5 nm. We concatenated spectra from the two channels using UV-2 for wavelengths $< 355$ nm and Vis for the remaining range. There was only a slight discontinuity in the joined spectra at the overlap wavelength. However, we found that use of the UV-2 wavelengths did not significantly improve results as compared with those using the Vis channel alone. Therefore, all results shown below are obtained using a range or subset of the range 355-502.5 nm obtained with the OMI Vis channel only. We note that there is a slight spatial mismatch between OMI UV-2 and Vis channels that may have contributed to the overall lack of improvement

280 using UV-2 wavelengths. In addition, there is limited high frequency spectral structure in the convolved $NO_2$ absorption cross sections in the UV-2 range (see Fig. 2). When PACE OCI data become available, we encourage testing again using all available wavelengths including those with wavelengths $> 502.5$ nm that are not available from OMI.

 Table 2 shows the results of testing with different wavelength ranges and inputs on a single OMI row (row 1, zero based) and with and without noise using the SNR models (Fig. 1). The use of the SNR models assuming no spatial binning results in

285 significant degradation for both OCI and GLIMR. For OCI, we find that use of more restricted wavelength ranges results in little or no significant degradation in the results as compared with the full range of 355–500 nm. Note that in all cases, the training target is SCD from the OMNO2 algorithm that corresponds to the OMNO2 fitting window, and all statistics are computed with respect to the OMNO2 SCDs. Our results are consistent with the full spectral resolution results of Li et al. (2022) who found an optimal retrieval window of 390–495 nm for estimating $NO_2$ vertical columns from TROPOMI radiances. Our results indicate

290 that most of the information for $NO_2$ at OCI spectral resolution is provided by the high frequency structure of the radiances produced by $NO_2$ absorption within the fitting window currently used in the OMNO2 product. Little additional information is provided by UV wavelengths that define the more broad $NO_2$ absorption feature. In addition, we show that removing the geometrical information (cosines of the solar zenith and relative azimuth angles) results in only a very small degradation. The use of an estimate of the stratospheric column $NO_2$ does appear to aid the estimation of the $NO_2$ SCD.

295 Most results in Table 2 are shown for 28 January 2005, a day not used in the training. For comparison, we also show results for a model with all predictors where we withheld data from 15 June 2005 from the training and instead used it for evaluation. On this day, the correlation is significantly lower as compared with 28 January 2005 and root mean squared difference (RMSD) slightly higher. In the northern hemisphere, there are high anthropogenic $NO_x$ emissions generally in populated regions. These emissions lead to higher $NO_2$ column amounts in the winter when lifetimes are generally longer. The solar zenith angles

300 are also higher in winter than in summer. These factors lead to higher SCDs in winter in the northern hemisphere populated regions than in summer. The higher $NO_2$ SCDs and variability in the northern hemisphere winter result in higher sensitivities and improved global statistics.

 The results for GLIMR shown in Table 2 were not satisfactory even without adding noise. GLIMR results were substantially worse with added noise, even after increasing the nominal SNR by factors of 3 and 6 that would be equivalent to averaging the

305 spectra of 9 and 36 pixels together, respectively. One reason for the poor performance is the lack of higher frequency spectral structure of the $NO_2$ effective cross sections at GLIMR resolution, particularly at blue wavelengths ($>400$ nm). Another factor is that there are many fewer available spectral samples for GLIMR, and therefore the impact of instrumental noise is substantial. No additional results will be shown for GLIMR.

 Table 3 shows results of the trained NN applied to data from all rows on 28 January 2005. Here, we report statistics for

310 SCDs normalized by the stratospheric AMF (essentially assumes that all $NO_2$ is in the stratosphere) to give a rough estimate of the VCD. Simulations without noise produced quite reasonable results, capturing about 94% of the variability with little overall bias. Results degrade noticeably when the SNR from Fig. 1b is applied to the simulated spectra; variability captured drops to about 91% and the RMSD increases from about 0.30 to 0.36 $\times 10^{15}$ molec cm$^{-2}$. More than half of the degradation that results from adding noise is recovered when the SNR is increased by a factor of 4. Such an increase in SNR could be

**Table 2.** Statistical comparison of SCD retrievals for row 1 simulated with the SNR models from Fig. 1 for OCI and GLIMR, using 17390 independent data points on 28 January 2005 (day not used in training, unless otherwise noted) and simulated OCI and GLIMR $NO_2$ SCDs. "All" refers to the use of all spectral data from the fitting window along with cosines of the solar zenith and relative azimuth angles (angs.) as well as the stratospheric column (strat. col.); statistics include the root mean squared difference (RMSD), bias, and variance explained ($r^2$) of the SCD. Bias and RMSD are given in units of $10^{15}$ molec cm$^{-2}$. Unless indicated under the noise column, all results with noise (indicated by 'Y') use the SNR with no assumed spatial binning as discussed in the text.

| inst | noise | NN inputs | fitting window (nm) | $r^2$ | bias | RMSD |
|------|-------|-----------|---------------------|-------|------|------|
| OCI | N | all | 355-500 | 0.964 | -0.131 | 0.805 |
| OCI | Y | all | 355-500 | 0.933 | -0.227 | 1.096 |
| OCI | Y | all | 400-500 | 0.937 | -0.141 | 1.040 |
| OCI | Y | all | 400-470 | 0.932 | -0.177 | 1.084 |
| OCI* | Y | all | 355-500 | 0.876 | -0.140 | 1.176 |
| OCI | Y | no angs. | 355-500 | 0.933 | -0.264 | 1.117 |
| OCI | Y | no angs., no strat. col. | 355-500 | 0.911 | -0.200 | 1.248 |
| OCI | Y | no strat. col. | 355-500 | 0.918 | -0.145 | 1.186 |
| GLIMR | N | all | 355-500 | 0.913 | -0.228 | 1.237 |
| GLIMR | Y | all | 355-500 | 0.885 | -0.283 | 1.435 |
| GLIMR | Y, SNR × 3 | all | 355-500 | 0.890 | -0.306 | 1.429 |
| GLIMR | Y, SNR × 6 | all | 355-500 | 0.912 | -0.242 | 1.254 |

*Data from 15 June 2005 withheld from training and used as evaluation (16286 samples)

achieved by averaging together spectral from a 4×4 array of OCI pixels to give an area of approximately $16\,km^2$ which is still an improvement over TROPOMI or averaging over 16 days of good observations. We computed statistics for a sample of data with less pollution ($NO_2 < 4 \times 10^{15}$ molec cm$^{-2}$). Here we see a significant decrease in correlation, likely due to lower variability within the sample and decreased sensitivity, consistent with results shown for a summer month in Table 2. The RMSD was also a bit smaller for this cleaner sample.

We also looked at how the performance varies across the OMI swath. We found better performance at the OMI swath edges where the SCD values are largest, owing to larger view angles, leading to deeper absorption features (last two lines of Table 3; swath edges are the rows with both the highest and lowest values). Even when normalized by the stratospheric AMF, the performance enhancement at the swath edges remains.

Figure 4 shows normalized SCD results obtained with data from 28 January 2005 using the PACE OCI SNR × 2 scenario.
The bulk of the normalized SCDs are in the approximate range of $1 - 4 \times 10^{15}$ molec cm$^{-2}$. Figure 4b shows that there is little evidence of striping (systematic row dependent errors) in the NN results. Note that there was no attempt to destripe the SCDs on this day of independent data as is done on a daily basis in the OMNO2 data used to train the NN. Striping can occur on an orbital or daily basis which is why OMNO2 initial SCDs undergo a destriping process. Figure 4c shows that there are some

**Table 3.** Statistics computed on on 28 January 2005 (day not used in training) for simulated PACE OCI slant column density normalized by stratospheric air mass factor with the SNR model from Fig. 1b. Statistics include the root mean squared difference (RMSD), bias, and variance explained ($r^2$) of the normalized SCD (normalized by the stratospheric air mass factor). Bias, RMSD, and thresholds are given in units of $10^{15}$ molec cm$^{-2}$. The number of samples was 1,113,061 except for the experiment labelled $NO_2 < 4$ that used 831,815 samples.

| instr. | noise | $r^2$ | bias | RMSD | area (km$^2$) |
|---|---|---|---|---|---|
| OCI no noise | N | 0.941 | -0.022 | 0.295 | 1 |
| OCI noise | Y | 0.911 | -0.045 | 0.363 | 1 |
| OCI SNR $\times$ 2 | Y | 0.925 | -0.037 | 0.333 | 4 |
| OCI SNR $\times$ 4 | Y | 0.933 | -0.029 | 0.315 | 16 |
| OCI SNR $\times$ 4, $NO_2 < 4$ | Y | 0.821 | -0.013 | 0.306 | 16 |
| OCI SNR $\times$ 4, rows 10-50 | Y | 0.923 | -0.033 | 0.338 | 16 |
| OCI SNR $\times$ 4, rows 0-10, 50-59 | Y | 0.953 | -0.022 | 0.262 | 16 |

systematic differences between OMNO2 and the NN normalized SCDs as a function of latitude. For example, the NN-derived
SCDs are lower than OMNO2 at low southern latitudes (over Antarctica) and also at middle northern latitudes that tend to occur for higher values of SCD (see Fig. 4a,d), likely over polluted regions. The NN values are somewhat higher near the equator. These systematic differences are not understood and may result from errors in either OMNO2 or the NN.

Figure 5 shows an example of how the $NO_2$ SCD errors may be spatially correlated. Because we use OMI data as the "truth" or target for training, simulated PACE OCI results are shown at the OMI spatial resolution rather than at the OCI resolution.
The reader must imagine that real OCI results can be obtained at a spatial resolution as high as $1\,km \times 1\,km$, rather than at the OMI resolution ($12\,km \times 24\,km$ at nadir) shown here. Judd et al. (2020) provided examples of fine resolution $NO_2$ over an urban area as measured from an aircraft instrument.

Here, we display retrieved SCDs based on simulations at various SNR levels on 28 January 2005, a day with high pollution over northeast and midwest portions of the United States. Figure 5a shows the original OMNO2 SCDs normalized by the
stratospheric AMFs to provide values similar to the total VCDs. These are considered as the true values used for comparisons with those from simulated OCI retrievals. Figure 5b shows the estimated fraction of radiance coming from cloudy portions of a pixel (cloud radiance fraction). In areas that have cloud radiance fractions approaching unity, most of the observed $NO_2$ SCD would result from $NO_2$ in the stratosphere or upper troposphere owing to clouds that screen the polluted atmosphere below them from satellite view. Note that pixels with snow cover that is not reported in the input snow/ice dataset will be reported as
cloud cover.

Results obtained using the simulated PACE OCI spectra with no noise added (Fig. 5c,d) show relatively small regional differences, with the largest differences occurring over highly polluted areas with some spatial dependence. Differences over polluted regions may occur as wavelengths in the UV, that OCI retrieval uses, have less sensitivity to $NO_2$ in the boundary layer where $NO_2$ can accumulate in high pollution conditions. It is also possible that there are imperfections in the OMNO2 data. The
"no noise" case represents the best result (upper limit) that can be expected based on the OCI sampling and spectral resolution

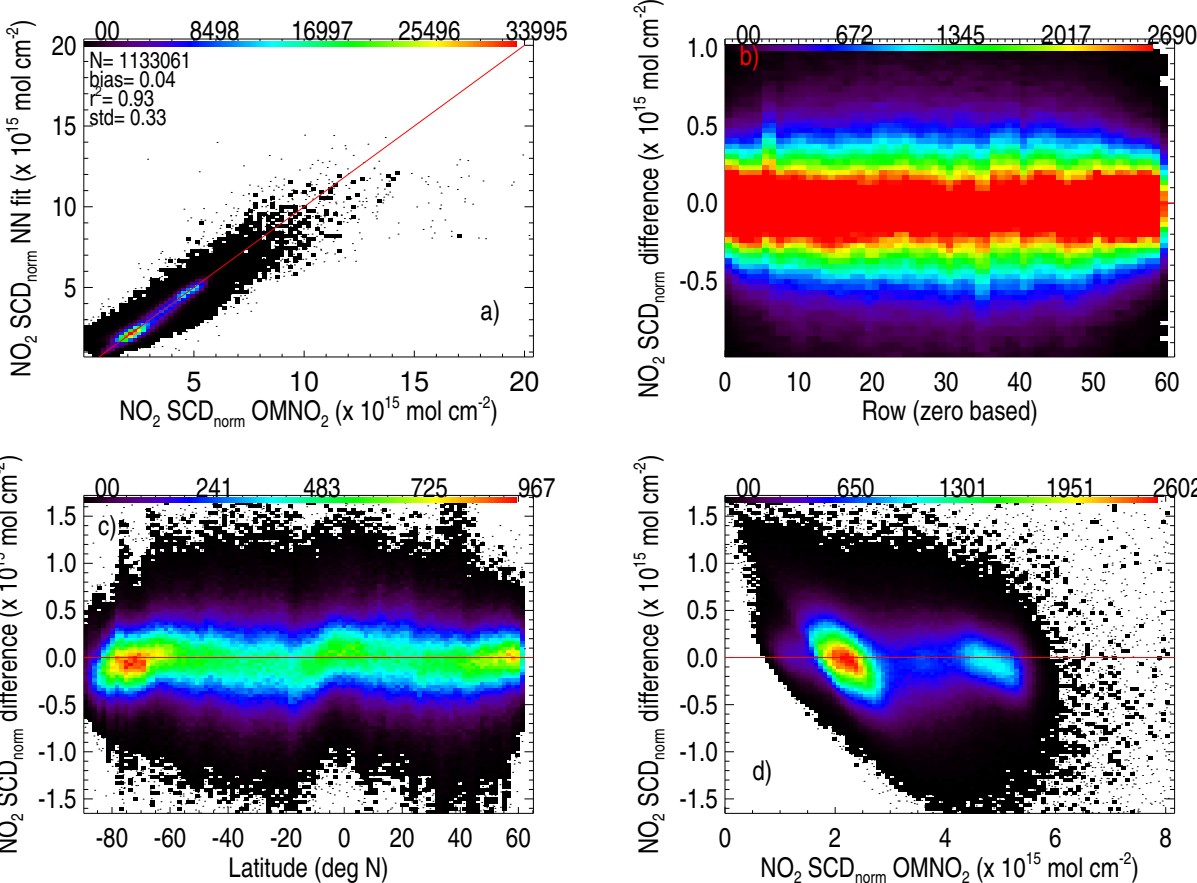

**Figure 4.** Global data from 28 February 2005 from a training using where each point represents the result that would be obtained using the OCI spectral characteristics and SNR from Fig. 1b multiplied by 2 with fitting window 355-500 nm; a) Density distribution (numbers along the top) of $NO_2$ target OMNO2 SCD (all results here are normalized (as indicated by the norm subscript) with respect to the stratospheric air mass factor) versus those from the NN training with statistics include standard deviation of the difference (std), fraction of variance explained ($r^2$), and mean of difference between NN and target (bias); b-d) density distribution of the $NO_2$ SCD difference (NN SCD - OMNO2 SCD) as a function of OMI row number, latitude, and $NO_2$ SCD, respectively.

for this particular training scenario. We tried alternate training scenarios such as training and applying NNs separately over land and water, but this failed to remove all of the differences.

The effects of the expected PACE OCI instrument noise, shown in Figure 5e and f, degrade the results with noticeably larger differences in $NO_2$ SCDs over the relatively clean oceanic regions and spatially dependent differences over polluted areas.
The effect of adding random noise to the spectra causes the neural network to draw less closely to the input data, and the ultimate effect may be to produce systematic or spatially dependent errors as well as random errors. Figure 6a-d shows results

of simulations using the OCI SNR multiplied by 2 or 4. The effect of increasing the SNR reduces the spatially dependent differences in polluted areas.

Figure 6e shows fitting uncertainties as given in the OMNO2 data set. The SCD uncertainties generally correspond to values of ~0.12 to 0.6 in the normalized SCD. For the case of no noise added to the simulated OCI spectral in Figure 5c and d, most differences fall within the range of fitting uncertainties. As noise is added to the simulated OCI spectra, the differences start to fall more outside the OMNO2 fitting uncertainties, particularly in the polluted areas. Figure 6f shows a histogram of normalized SCD for low values typical of cleaner regions ($0$-$4 \times 10^{15} \, \mathrm{mol \, cm^{-2}}$) for OMNO2 and for the case of OCI spectra with noise added according to $2 \times$ the SNR model. Here we see a more peaked distribution of values for the NN estimates. This may indicate that the NN with inputs of leading PCA coefficients may be reducing the effects of random instrument noise. We further investigate potential noise reduction for a cleaner region in Sect. 3.3.

## 3.2 Practical implementation issues

We next address how our approach can be practically implemented with a high spatial resolution, low spectral resolution hyperspectral sensor such as PACE OCI and an existing moderate spectral resolution spectrometer such as TROPOMI. The desired retrieved quantity for atmospheric correction in ocean color algorithms is not the $NO_2$ SCD for a particular fitting window, but rather the VCD such that the appropriate absorption can then be accurately computed at any wavelength for atmospheric correction (Ahmad et al., 2007). Once a NN is trained to produce $NO_2$ SCD from input spectra, the SCDs may be converted to VCD using a computed AMF that will be a function of the sun-satellite geometry, surface and cloud conditions, and $NO_2$ profile shape as described above and shown in Figures 7-8. This is typically accomplished with lookup tables and model-generated $NO_2$ profiles. With the $NO_2$ VCD, the spectral transmittance due to $NO_2$ can then be computed with a radiative transfer model that will be a function of wavelength, the surface albedo, and other absorbers and scatters in the atmosphere. This last step may be performed with either table lookup or machine learning.

To use $NO_2$ SCDs from TROPOMI or similar spectrometer for training of a NN with a collocated spectra from a higher spatial resolution hyperspectral instrument, such as OCI, as inputs for the purpose of estimating SCDs from the imager, it is necessary to use radiative transfer to transform the SCDs from the spectrometer to the appropriate sun-satellite geometry of the imager as shown in Figure 7; the overlap between these two instruments on different satellites will not typically occur for the same sun-satellite geometries. One approach to prepare a training data set would be to use the total VCD retrievals from the spectrometer (e.g., TROPOMI) to derive an estimated SCD for the imager (e.g., OCI) at its geometry. One can simply apply computed AMFs at the appropriate geometries. This can be written as

$$\mathrm{SCD_{imager}} = \mathrm{SCD_{spectrometer}}/\mathrm{AMF_{spectrometer}} \times \mathrm{AMF_{imager}} = \mathrm{VCD_{spectrometer}} \times \mathrm{AMF_{imager}}, \qquad (1)$$

where $\mathrm{AMF_{imager}}$ is the AMF computed using the same inputs as for the spectrometer (e.g., cloud radiance fraction, $NO_2$ profile shape, etc.) but using the appropriate sun-satellite geometry, including its impact on surface bi-directional reflectance (currently accomplished using the GLER framework), for the imager observation. This kind of approach could be made to work

relatively quickly with mature and well validated VCDs from a spectrometer without need to understand or make adjustments for instrumental artifacts in the imager.

The imager spatial resolution will be higher than that of the spectrometer. One way to prepare a collocated training set would involve averaging the spectra from the imager to match the footprints of the spectrometer. This averaging will effectively change the SNR of the imager data as compared with that at its native spatial resolution. The resulting training will therefore not be optimized for application at the native resolution. If the trained network is then applied at the native resolution of the imager, the results will have to be carefully validated and checked. Another possible approach would be to spatially interpolate the spectrometer data to the locations of the imager or to perhaps use a high resolution model to downscale the spectrometer data to the resolution of the imager. Addressing these details is beyond the scope of the present work and will be dealt with in future studies.

One advantage of using collocated data from a spectrometer is that a fitting algorithm would not have to be developed and tested for the imager. We have found that developing and validating such algorithm can require significant human resources. However, our results suggest that it is possible to develop a fitting algorithm for the imager by exploiting the high frequency structure of $NO_2$ absorption. Such an approach, not requiring collocated data from another instrument, can be considered as an alternative approach. Spectral fitting algorithms can be computationally intensive and it may still be desirable to use machine learning to speed up the processing of dense imager data as shown in Figure 8. For example, Li et al. (2022) found that a NN implementation for $NO_2$ vertical columns using TROPOMI spectra was about 12 times faster than a full implementation using a priori profiles from a high spatial resolution chemistry-transport model.

Another consideration is how often a NN would need to be retrained. If the instrument were spectrally stable, retraining might not be necessary or might be infrequent. However, destriping may still be necessary to correct for transient spectral artifacts. Retraining should be done whenever there is a substantial change in the instrument spectral characteristics. Since the OMNO2 algorithm uses monthly-averaged solar irradiances, it may be more optimal to similarly normalize with respect to the same set of solar irradiances before training than to use only the radiances as we have done here as the solar data may help to account for instrumental changes.

### 3.3 Use of PCA and a neural network for reducing noise in $NO_2$ slant column retrievals

We next explore the use of PCA in conjunction with a NN for noise reduction in SCD retrievals. In addition, a NN implementation may have the benefit of significantly reducing computation time of the spectral fitting algorithm. Machine learning that uses leading PCA coefficients as inputs may be adept at filtering out spectral interferences as well as random instrument noise.

For the following experiments with real OMI data, our assumption is that over a generally clean environment (Pacific ocean), variability in $NO_2$ SCD, due to clouds for example, is relatively small. In this region, the majority of the $NO_2$ column resides in the stratosphere and upper troposphere, where UV wavelengths have good sensitivity. We use the same training days as above and evaluate using data from an independent day (28 January 2005).

Table 4 shows the standard deviation of normalized SCDs computed over the region bounded by -10S to 20N latitude and 110E to 173E longitude. for OMNO2 and three separately trained NNs. Here, we attempt to disentangle the impact of three

**Table 4.** Standard deviation for $NO_2$ slant column density normalized by stratospheric air mass factor of the Pacific region shown in Fig. 9 ($\sigma_{\mathrm{Pacific}}$) over 40780 points on 28 January 2005 (day not used in training) in units of $10^{15}$ molec cm$^{-2}$.

| Data Set | Fitting Window (nm) | spec. res. (nm) | $\sigma_{\mathrm{Pacific}}$ |
|---|---|---|---|
| OMNO2 | 402–465 | 0.63 | 0.43 |
| NN trained | 402–465 | 0.63 | 0.28 |
| NN trained | 355–500 | 0.63 | 0.30 |
| NN trained | 355–500 | 5.0 | 0.28 |

different factors on the retrieval noise: 1) use of a NN with leading PCA coefficients as inputs and a spectral fitting window mirroring that of OMNO2; 2) NN trained similar to 1) but with an extended spectral range; and 3) a NN similar to 2) but spectrally averaged to coarser resolution. The results of using the NN with PCA approach with the OMNO2 spectral fitting window (402–465 nm) and at OMI spectral resolution decrease in variability in the region by about 30%. Nearly identical reductions are obtained using the NN-PCA set up with either an increase spectral fitting windows (355–500 nm) or reduced spectral resolution (using 5 nm as compared with the OMI native resolution of 0.63 nm). The results of these experiments suggest that it is the NN set up with leading coefficients of PCs that is leading to the noise reduction. The NN approach combined with PCA appears to be effective in isolating information about $NO_2$ in the spectra while rejecting interfering spectral features and random noise.

Figure 9a,b shows the noise reduction visually using NN results obtained with the OMNO2 spectral fitting window (402–465 nm) for a day not used in the training. There appears to be a noticeable reduction in random noise over this clean region. The noise reduction is particularly apparent over highly cloudy regions (see Fig. 9c) where the atmosphere below the clouds is shielded from satellite view and the majority of the observed SCD is in the stratosphere. The difference map in Fig. 9d shows mostly random patterns with magnitudes of similar magnitude to the OMNO2 fitting uncertainties shown in Fig. 9e. Figure 9f shows a histogram of the retrieved normalized SCDs from the target OMNO2 and the NN. OMNO2 has more pixels in the tails in the distribution, particularly at the low end. Negative SCDs are reported in the OMNO2 data set. These are retained for statistical purposes although they are physically unrealistic. The trained NN substantially reduces these low values and also reduces the number of pixels at the high SCD tail of the distribution. Similar results are obtained with the other setups described above (lower spectral resolution and a wider spectral fitting window).

## 4 Conclusions

We have simulated data from the hyper-spectral imagers PACE OCI and GLIMR using OMI to demonstrate that it is possible to retrieve $NO_2$ SCD with reasonable accuracy and precision using lower spectral resolution data from the PACE OCI. Instrumental noise significantly impacts the results as does the spectral resolution and sampling. Better results are obtained in cases of $NO_2$ pollution contained in the boundary layer when the spectral resolution is high enough (of the order of 5 nm or better) to capture the higher frequency spectral structures in the blue part of the $NO_2$ absorption spectrum. The longest OMI

wavelength is at about 500 nm; OCI spectral coverage will continue to longer wavelengths in the green and yellow parts of the spectrum where $NO_2$ has additional absorption features. The added spectral range may improve results over those shown here,

although the magnitude of improvement is not expected to be dramatic. Averaging spectra from several adjacent OCI pixels together will improve performance of $NO_2$ retrievals from PACE OCI that may be used for atmospheric correction in ocean color retrievals.

There are other potential applications for PACE OCI $NO_2$ retrievals. Currently, $NO_2$ tropospheric column density retrievals from instruments such as OMI and TROPOMI are averaged over various time periods to reduce the impacts of retrieval noise

and meteorology (e.g., Lamsal et al., 2015; Duncan et al., 2016). Similar averaging of $NO_2$ data from imagers over time, e.g., of the order of a month or more, may produce good quality data at higher spatial resolution than is available from TROPOMI. This higher spatial resolution data could then be used to downscale TROPOMI and historical OMI retrievals or could be used for emissions estimates based on averaged maps, for example using recently developed methods for high resolution data (Liu et al., 2022). Use of high resolution averages is also useful for studies involving health impacts, including investigations

involving environmental inequities (e.g., Goldberg et al., 2021; Kerr et al., 2021; Cooper et al., 2022).

We also show that our machine learning with PCA approach for OCI can be used to reduce noise in retrieved $NO_2$ SCDs (at the least in unpolluted situations) for spectrometers such as OMI and TROPOMI. An additional advantage of using machine learning for noise reduction in spectral fitting is that once trained, an applied neural network is an extremely efficient algorithm. The current OMNO2 spectral fit is the most computationally intensive portion of the OMNO2 $NO_2$ tropospheric column

retrieval algorithm. This may be an important consideration with the new generation of sensors in geostationary orbit with very large data volumes (Zoogman et al., 2017). These include the Korean Geostationary Environment Monitoring Spectrometer (GEMS), NASA Tropospheric emissions: Monitoring of pollution (TEMPO), and Copernicus Sentinel 4. Training could be applied intermittently throughout the data record to ensure that time-varying instrumental artifacts are accounted for. We stress that a high quality physically-based fitting algorithm is a necessary part of any machine learning approach as it produces the

retrievals needed as the training target. Our machine learning approach is not meant to replace these algorithms, but rather to enhance and speed them up.

*Data availability.* OMI level 1B and level 2 $NO_2$ products used in this study are available from the GES-DISC at https://disc.gsfc.nasa.gov as cited in the manuscript.

## Appendix A: Typical radiance distributions

Figure A1 shows typical radiance distributions for OMI row 1 taken over the same orbits as used for the neural network training as described above. The radiance distribution depends upon how the incoming solar irradiance is modified by gaseous and particle scattering and absorption in the atmosphere as well as surface reflectance properties.

*Author contributions.* JJ was responsible for the design of the methodology, investigation, including software, visualization, calculations, and formal analysis, and wrote the first draft of the article. JJ, AV, and NK provided overall supervision of the project. NK and JJ contributed to funding acquisition. LL, NK, ZF, CL, and SM provided guidance on the use of OMI products. All authors contributed to article revision, read, and approved the submitted version.

*Competing interests.* The authors declare that they have no conflict of interests.

*Acknowledgements.* The authors thank the international OMI team that produced and distributed the OMI data sets used here. They also thank the PACE OCI and GLIMR teams, particularly Antonio Mannino, Bryan Franz, Shihyan Lee, Amir Ibrahim, Andrew Sayer, Brian Cairns, and Gerhard Meister who provided the signal-to-noise (SNR) estimates used here and assistance in simulating PACE OCI and GLIMR data. Finally, the lead author thanks Arlindo da Silva for enlightening conversations. This work was supported by NASA through the PACE and Aura science team programs.

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

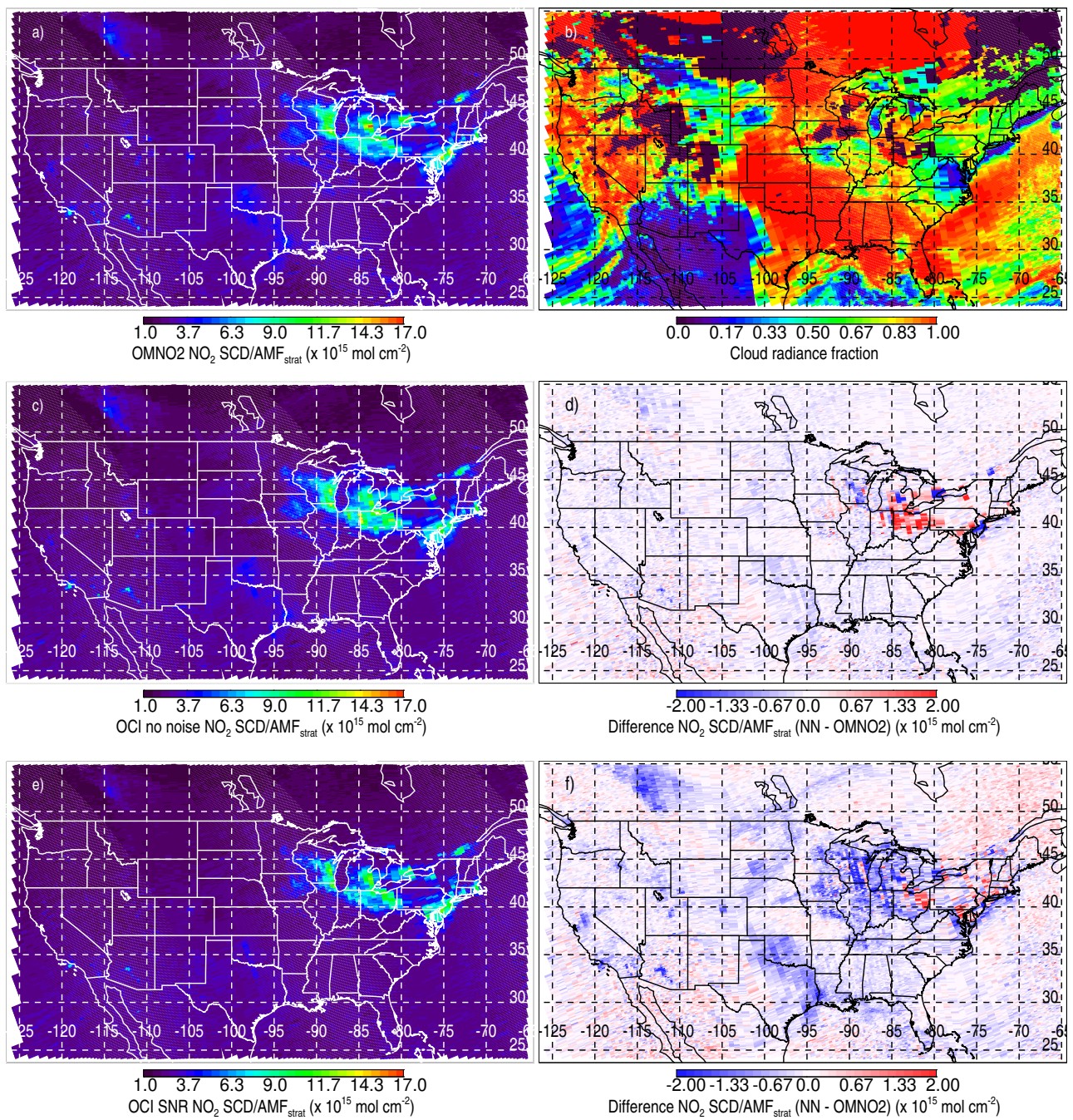

**Figure 5.** Data from 28 January 2005 (day not used in fitting) for $NO_2$ SCDs normalized by the stratospheric air mass factor (normalized SCD); 2) Normalized SCDs from OMI OMNO2 algorithm; b) Cloud radiance fraction for the $NO_2$ fitting window from OMNO2; c), e): Normalized SCDs from the NN algorithm evaluated at each OMI pixel using OCI spectral characteristics and fitting window of 355–500 nm with no noise and with the SNR model of Fig. 1b, respectively; d), f): corresponding differences with respect to OMNO2 normalized SCDs for the results shown in (c) and (e), respectively.

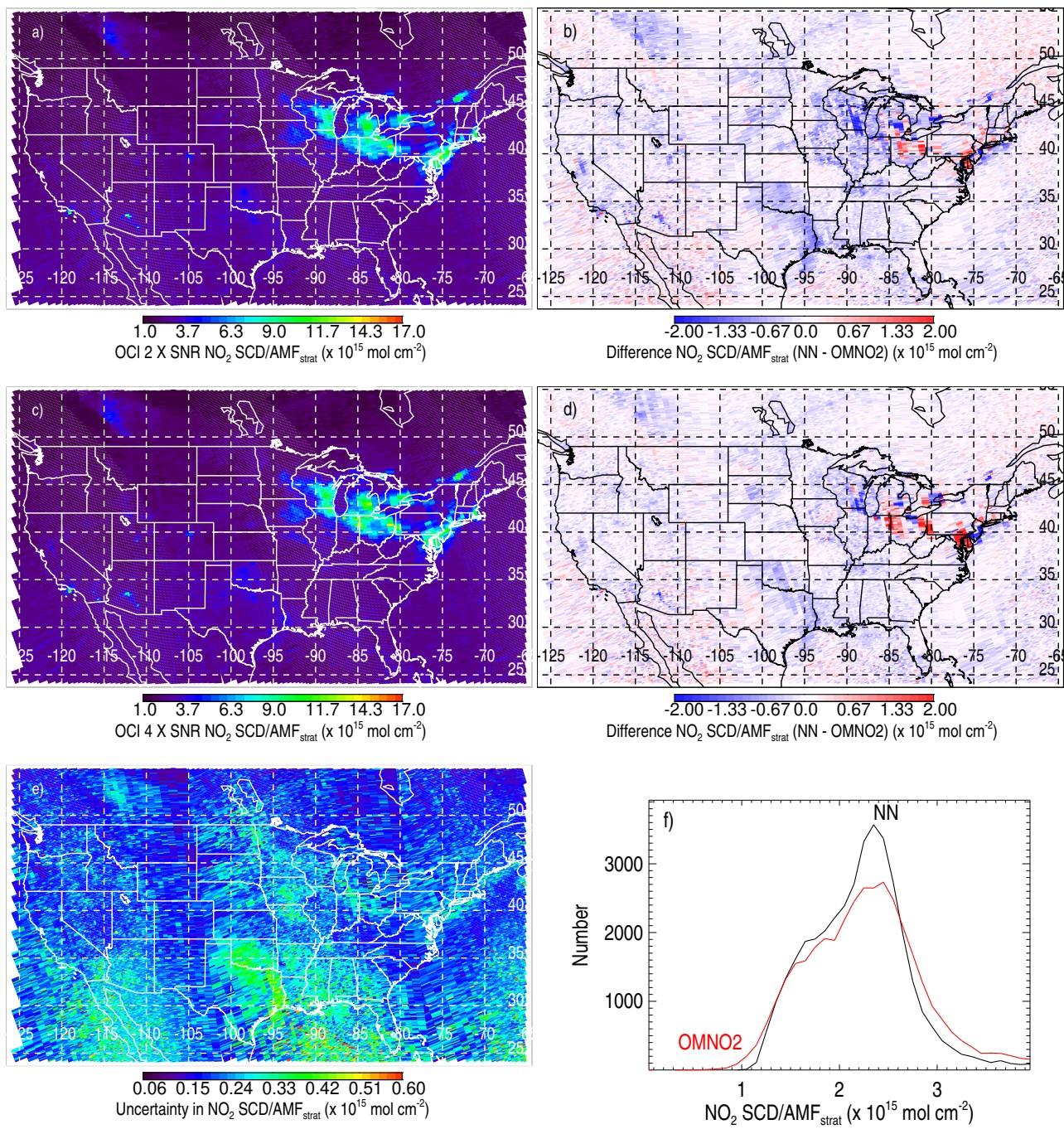

**Figure 6.** Similar to Fig. 5 showing data from 28 January 2005 (day not used in fitting): a), c) $NO_2$ SCDs normalized by the stratospheric air mass factor (normalized SCD) and for NN algorithms trained on data sets using OCI spectral characteristics and fitting window of 355–500 nm, evaluated at each OMI pixel using $2\times$ and $4\times$ the SNR from the model of Fig. 1b, respectively; b), d): corresponding differences with respect to OMNO2 normalized SCDs shown in Fig. 5a.; e) fitting uncertainties in normalized SCD from OMNO2; f) histograms of the lower end of normalized SCDs from OMNO2 and the NN model with $2\times$ the OCI SNR.

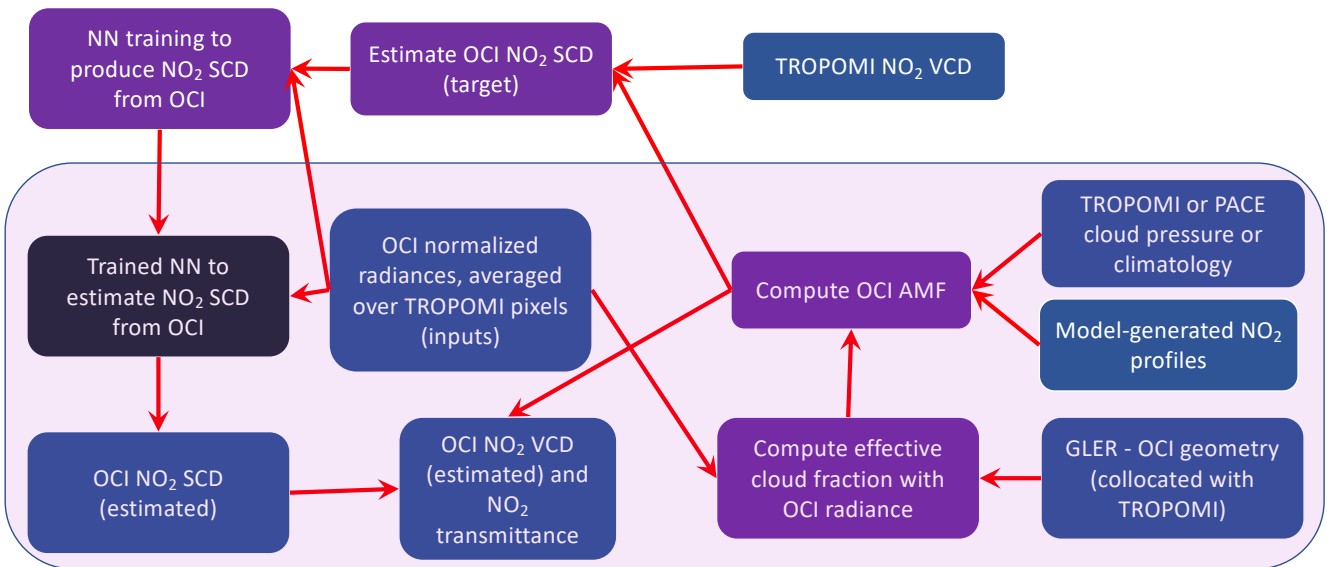

**Figure 7.** Flow diagram showing processes and data needed to estimate the $NO_2$ VCD using TROPOMI data collocated to OCI. Data sets are indicated by blue boxes, dynamic processes with purple, and a static process in black. Components needed to compute the VCD once the neural NN training process is complete are shown within the light purple overlay.

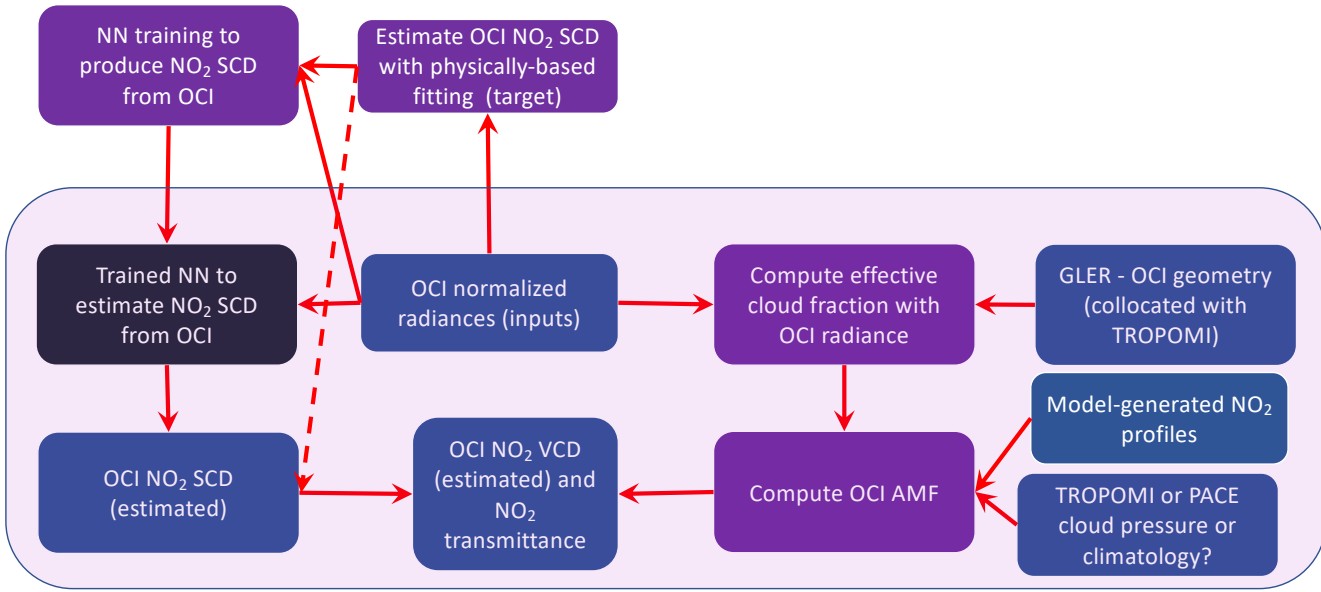

**Figure 8.** Similar to Figure 7 but showing processes to estimate $NO_2$ VCD using only OCI data. The dashed line shows an alternative flow that does not require training a neural network.

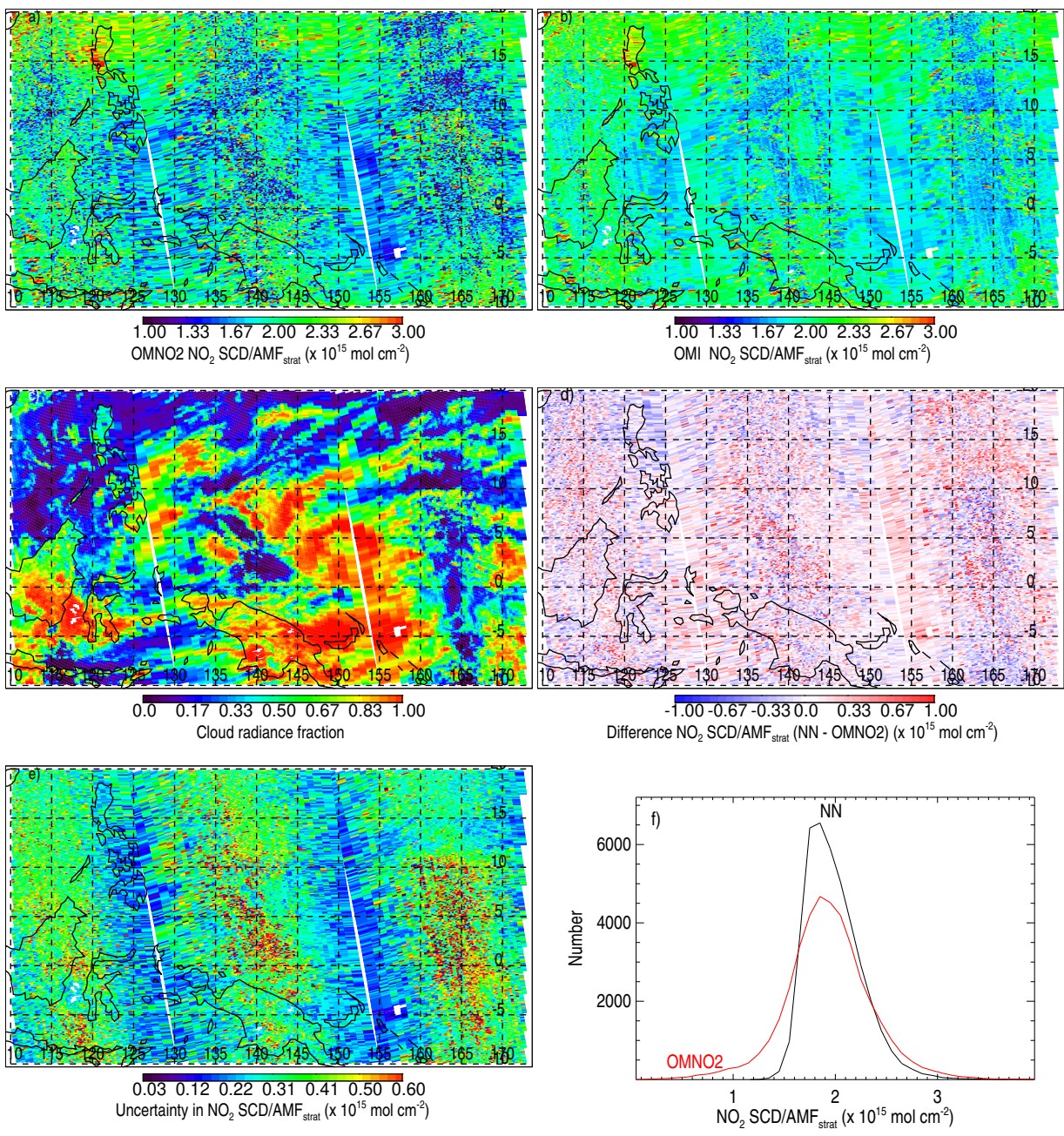

**Figure 9.** Normalized NO$_2$ SCDs from 28 January 2005 over the tropical Pacific region from a) OMNO2 and b) NN applied at OMI spectral resolution and with the OMNO2 fitting window 402-465 nm; c) cloud radiance fraction; d) difference between the NN and OMNO2 normalized SCDs; e) normalized SCD fitting uncertainties from OMNO2; f) histograms of the normalized SCDs corresponding to the area shown in (a).

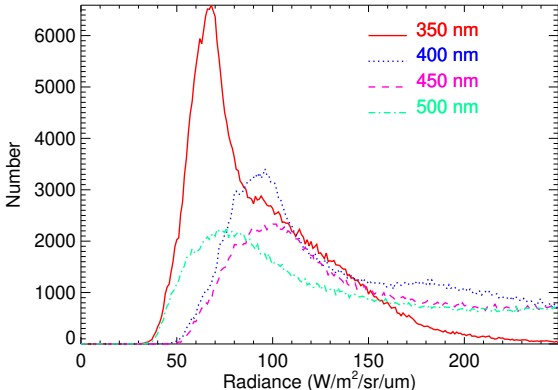

**Figure A1.** Histograms of radiance at different wavelengths from OMI row 1 (zero based) data taken over a range of orbits as described above.