# Peer review of "Use of machine learning and principal component analysis to retrieve nitrogen dioxide $(NO_2)$ with hyperspectral imagers and reduce noise in spectral fitting"

_EGUsphere, 2022_

## Author Response (AR1)

Dear Dr. Van Roozendael:

**Below we respond to each of the reviewer's comments in bold and show changes to the manuscript in blue bold font. In addition, as we reorganized the paper and recomputed all results using additional training days, we tightened up the manuscript in terms of the text. All changes are reflected in the marked up version.**

**Sincerely,**

**Joanna Joiner**

==============================================================

**Response to RC1:**

**We thank the reviewer for their careful review that has helped to improve the manuscript and reply to each comment (repeated below) in bold face.**

The paper by Joiner et al. is a carefully, well written paper. It contains interesting results. First of all, it shows that a meaningful retrieval of NO2 is possible at a moderate spectral resolution of about 5 nm as provided by OCI. Secondly, it demonstrates a neural net PCA approach as a fast way to do the retrievals, an approach with interesting properties. As such I am in favour of publishing these results. However, I have several questions I would ask the authors to address in the final version of the paper.

page 5, line 108: "The conversion of SCD to total or tropospheric VCD can be accomplished in a straight-forward and computationally efficient manner as in current algorithms and is not addressed further here. " Nevertheless, I understand that the largest contributions to the overall error is in the conversion to vertical columns. Information on the clouds, aerosols and albedo is important for this step. It would be of interest to comment if such information is available from OCI/PACE.

**Such information will be available from OCI/PACE. We add this to the discussion: "Information about the surface, aerosol, and cloud, such as cloud/aerosol radiance fraction and effective pressure needed for calculation of the AMF (e.g., from the $O_2$ A band), will be available from OCI itself (Werdell et al., 2019)."**

line 158: "independent of radiance value." The cloud-free scenes are the most relevant ones, and these have lower radiance levels than cloud covered scenes. Does this assumption not lead to an over-optimistic representation of the results?

**We do not believe that the first SNR model we used (independent of radiance value) produces over-optimistic results for OCI. Compared to the second more refined SNR model based on more recent measurements, the SNRs for the radiance-independent model are overall lower, except when radiances values drop below 50 W/m$^2$/sr/um. We included PDFs of OMI measured radiance in the appendix (Figure A1) that show that for all wavelengths, only a small fraction of scenes has radiances below 50 W/m$^2$/sr/um. The radiance-independent SNR model may be in fact pessimistic as shown by the comparisons with the more refined SNR model (see Figs. 5-6 and Table 2 in the revised manuscript and those in our original manuscript). To avoid confusion, we have removed results with the OCI radiance-independent SNR model and now show only results from the most recent radiance-dependent SNR model. For GLIMR, we have only a radiance-independent model, but we also now include the most optimistic results for GLIMR with no noise added. In addition, we added "Note that for determination of NO$_2$ tropospheric vertical columns, cloudy observations are used to help estimate stratospheric column amounts, so that the full range of radiance values is needed, not just clear-sky observations (Bucsela et al., 2013)."**

line 172: "The PCA concentrates the spectral features" -> The PCA concentrates on the spectral features

**fixed, thank you.**

line 275 "nomainal"

**fixed, thank you.**

line 280: "GLIMR, even with no noise, were not satisfactory .. are not shown". I would like to ask the authors to consider to provide them anyway, or add a few lines to table 1 or 2. Linked to figure 1, this would more explicitly show that meaningful retrievals are possible up to 5nm resolution, but that 10nm is removing much of the useful NO2 information.

**We provided GLIMR results with and without noise in Table 2 (the old Table 1) as requested and added text to go along with it.**

How does the NN-OMNO2 RMS compare to the OMNO2 retrieval uncertainty?

**We have provided maps of the fitting uncertainties from the OMNO2 retrieval in the revised manuscript (Fig. 6e and Fig. 9e) with explanatory text. In comparison with the difference maps, these new uncertainty maps now provide context. In areas of low pollution, the differences are of the same order as the fitting**

**uncertainties. However, in polluted regions, the differences may be larger than the uncertainties.**

It seems there is still quite a spread and some systematic effects caused by the NN approach (Fig.4). Please comment and put the results in perspective.

**Instead of showing results with the nominal SNR model in Fig. 4, we now show results with the more refined SNR model assuming 4 pixels are averaged (as we expect to do averaging over time and/or space). We also now show the results in terms of normalized slant column (close to the total column) rather than slant column as researchers are more familiar with this quantity. As noted, there is still significant spread and systematic effects (also seen in the figures with maps). The addition of the fitting uncertainties to the figures with maps helps to put the results in perspective as discussed on the previous point.**

Data is shown for 28 January over the US, when both the solar angle and NO2 column amount is relatively high. It would be interesting to show also an example (maybe in the tables only) for the summer to check the seasonality of the differences.

**We now provide results in Table 1 (now Table 2) for a summer date as suggested where we removed this date from the training and used it for independent evaluation. In looking at the initial results, we decided to add more training data, so all results have been recomputed using one day each month instead of every other month (plus additional high pollution days) and choosing every third sample. We added a paragraph to explain the results: "Most results in Table 2 are shown for 28 January 2005, a day not used in the training. For comparison, we also show results for a model with all predictors where we withheld data from 15 June 2005 from the training and instead used it for evaluation. On this day, the correlation is significantly lower as compared with 28 January 2005 and root mean squared difference (RMSD) slightly higher. In the northern hemisphere, there are high anthropogenic NO$_x$ emissions generally in populated regions. These emissions lead to higher NO$_2$ column amounts in the winter when lifetimes are generally longer. The solar zenith angles are also higher in winter than in summer. These factors lead to higher SCDs in winter in the northern hemisphere populated regions than in summer. The higher NO$_2$ SCDs and variability in the northern hemisphere winter result in higher sensitivities and improved global statistics." We added new results also in Table 2 for the winter month, but for a sample in clean air and saw a similar drop in correlation.**

line 313: "generally low bias over highly polluted areas" What is the reason for this low bias? I would expect instrument noise to lead to random effects, not a systematic bias.

**There may be systematic errors in the training data that lead the neural network to reduce the fitting errors in a certain way that may lead to spatially dependent differences. It is possible that a separate training could be done and applied for cases of high pollution if we are to completely trust the OMNO2 results. We revised this paragraph to replace the words "error" and "bias" with "differences" as OMNO2 used as the target of the training may not represent the absolute truth. We expanded the mapped area over the US to more clearly show both positive and negative differences that can be spatially correlated. We also added a histogram of the low column amounts over the mapped area that encompasses the US to illustrate a possible noise reduction for these cases as a preview of the additional analysis we conduct later for the tropical Pacific. We added text to describe the additional panels. We also added, "The effect of adding random noise to the spectra causes the neural network to draw less closely to the input data, and the ultimate effect may be to produce systematic or spatially dependent errors as well as random errors."**

line 364: "convolved with a 1 nm boxcar function" Could you explain why this is done, instead of using OMI radiances at their spectral resolution?

**To assure that we weren't fitting to the noise, we chose to do this convolution. However, the use of leading PCA coefficients should ensure that we are not fitting the noise. As stated in the response to the next comment, we have revised this part to remove the convolution.**

Figure 9: It was confusing to me what can be concluded from these plots. Several things were changed: wider spectral range, change of spectral resolution, NN versus standard retrieval. Is the effect due to the wider window, or could the NN procedure cause a lowering of the apparent noise? Is the NN distribution of SCD values more realistic than the OMNO2 one?

**We thank the reviewer for this question. To make this aspect clearer, we replaced the previous results with new experiments, now using exactly the same training days as for the OCI simulations. We try to disentangle the impact of using the NN with PCA coefficients as inputs as well as increasing the spectral range and decreasing the spectral resolution. We don't know what the true SCD values or distributions are but based on visual analysis in the clean region shown in Figure 9, we believe that we have reduced the effects of random noise in the original spectra that produced random noise in the retrievals. For example, we have reduced what is believed to be erroneously low values in the lower tail of the OMNO2 distribution (for example, the unphysical negative values). We added a table (Table 4 in the revised manuscript) that shows the standard deviation of**

**normalized SCDs over the mapped region for the different experiments and reworked this section.** The new experiments suggest that it is not the increased wavelength range that produces lower standard deviations or even the spectral resolution, but rather the approach of using leading PCA coefficients as inputs to the NN.

line 403: "could be used for emissions estimates". Most emission estimation methods make use of daily data, basically identifying the plume from a localised source, which then allows the estimation of the source (and these daily values may then be averaged in time). Would monthly-averaged maps be of use for emission estimates?

**Yes, averaged maps can be used for emission estimates using for example the method of Liu et al. (2022) as referenced rather than averaging emissions estimates from individual days. We added "based on averaged maps" here to make this clearer.**

=====================================================================

**Response to RC2:**

**We thank the reviewer for their careful review that has helped to improve the manuscript and reply to each comment (repeated below) in bold face.**

Joiner et al. presented a new method based on machine learning to retrieve NO2 from low spectral resolution instruments. This is an interesting study and the authors demonstrated well the potential of this approach for forthcoming missions with high spatial resolution. The paper is very well written and structured and there is little to say on the content. It should be published in AMT, after few (mostly minor) corrections.

Overall, the quality of the figures is quite poor, and it would be needed to improve the resolution of all figures.

**We tried to improve the quality of the figures. We think the issue was in conversion from .eps to another that format would work within the LaTeX template. We have tried another method that does not degrade the quality of the figures so that the text and content are sharper. Note: the flow diagrams were saved directly in pdf format, so they have not been altered.**

Comments

My main reservation is that the paper covers two things: (1) the retrieval of NO2 from instruments with lower spectral resolution, (2) the retrieval of NO2 using machine

learning. What I would like to see is a DOAS-type retrieval of NO2 for the lower spectral resolution data so that one can evaluate the benefit of the machine learning approach directly. I realize that this is probably quite some work but it would be nice to understand if machine learning can improve the retrievals or not. It is understood that machine learning is interesting in terms of computational time but from the results shown here it is not clear if it is the only advantage.

**We thank the reviewer for this question.** **We have modified the last paragraph of the introduction to make it clearer that we attempt to answer two questions related to the two points above. We also changed the title to reflect these two aspects.**

**We agree that it would be quite some work to do the DOAS-type retrieval and we were after a timely answer to the question of whether or not we could retrieve NO$_2$ with the lower resolution OCI sensor before its launch, so we turned to machine learning. We did find relevant references on DOAS type retrievals using the APEX airborne sensor whose resolution is ~2.4-3.4 nm across the NO$_2$ fitting window and have included them in the revised manuscript. They were only able to use a limited fitting window of 470–510 nm for DOAS NO$_2$ retrievals due to "interference with unidentified instrumental artefacts or features prevents us from extending the fitting window to wavelengths lower than 470 nm," so such retrievals appear to be difficult and beyond the scope of the present work.**

**Please see answers below for further clarification on the potential benefits of the neural network approach.**

-Section 2:

*A small section introducing OCI and GLIMR is missing here. Details on instruments (spectral range, sampling, performance, etc) should be added (e.g., as a form of a table).

**We have added such a table (Table 1 in the revised manuscript) with instrument characteristics as suggested and moved the SNR figures and some other text to this section.**

*It would be good to refer to past studies of low spectral resolution NO2 retrievals attempts.

**We found a few works with lower spectral resolution NO$_2$ retrievals: the APEX aircraft instrument (Tack et al., 2017; Kuhlmann et al., 2022) and a satellite study (Postylyakov et al., 2017, 2019; Zakharova et al., 2021) and have referenced them as appropriate.** We hope we haven't missed any other references.

*line 89 reads '..a retrieval would likely need to make use of the broad continuum absorption..'. Not sure what is meant here. Is the author meaning a large wavelength range to better constrain the fit or is it really the broadband contribution of the NO2 absorption which is targeted? Please clarify.

**We changed this sentence to "At GLIMR spectral resolution, a retrieval would likely need to make use of the broad NO$_2$ absorption feature peaking at around 400 nm rather than the finer spectral features used in DOAS retrievals."**

*p8, line 179 '...to half the number of spectral elements was sufficient to capture the spectral information associated with NO2 while providing some noise reduction'. I don't understand why this should provide a noise reduction. Could you elaborate?

**We have added text to elaborate: "Since the trailing PCs typically express random spectral noise, eliminating these modes can lead to noise reduction."**

*p9, 275: the use of the logarithm of the NO2 SCD does not correspond to anything physical. Could you clarify why this was used? Is there a justification for this, other than it gives good results.

**We added ", likely because the distribution of SCDs is more normally distributed in log space, which is desirable for neural network training."**

*p10, line 237: about the NN ability to capture the wavelength dependence of the SNR. What about wavelengths cross-correlations? It is this information available from the OCI and GLIMR? I guess not but could it affect the performance of the NN approach?

**We added "Here, we assume that errors are not correlated with wavelength as Information on correlated errors was not provided. Correlated errors could possibly degrade the performance of the retrieval if the neural network is not able to effectively account for them."**

-Section 3

**\*** for figs 5,6,9, it is not always clear to what settings they correspond. E.g., what wavelength range was used, noise added or not, etc. I propose to detail this in all figure captions.

**Details are now listed in the figure captions as suggested.**

* Table 1: why is the bias larger for the case where the wavelength range is close to that of OMI NO2 (400-470 nm)? In general, the SCDs in different windows should not be

identical (because of the AMF wavelength dependence). I find this aspect is not discussed enough.

**While we are using different wavelength ranges for the NN training, the training target output is the OMNO2 slant columns so that the output slant columns always pertain to the OMNO2 fitting window. The differences in biases between the different runs are fairly small (insignificant) and are probably related to uncertainty in the training. We added "Note that in all cases, the training target is SCD from the OMNO2 algorithm that corresponds to the OMNO2 fitting window, and all statistics are computed with respect to the OMNO2 SCDs."**

*P14, l 313: it is mentioned that the 'largest errors occur over highly polluted areas'. Could it be because the NN is not sufficiently trained for these conditions? Could the situation improve by adding such highly polluted scenes in the training set? If yes, what would be the weight of such scenes in the training set?

**As described in the section "Simulated OCI and GLIMR data", we added extra days with high pollution to provide more samples under these conditions and this improved the results a bit as compared with not including the extra days. Any imperfections in the OMNO2 data could also contribute, and these may be amplified in polluted conditions. We tried many different approaches to try to improve the results, e.g., separate training for land and ocean. We were not able to improve the results. It is possible that a separate training could be done and applied for cases of high pollution if we are to completely trust the OMNO2 results. We revised this paragraph to replace the words "error" and "bias" with "differences" as OMNO2 may not represent the absolute truth. We added a sentence discussing that we tested other training schemes: "We tried alternate training scenarios such as training and applying NNs separately over land and water, but this failed to remove all of the differences."**

*p14,l 328 : '..but rather the VCD'. Why? SCD is the actual signal, not the VCD.

**We modified the sentence to "The desired retrieved quantity for atmospheric correction in ocean color algorithms is not the $NO_2$ SCD for a particular fitting window, but rather the VCD such that the appropriate absorption can then be accurately computed at any wavelength for atmospheric correction (Ahmad et al., 2007)."**

* In general, how frequent should the NN be trained? Have you tried the algorithm for periods affected by the row anomaly of older data in the OMI mission? How is the data quality affected?

**We did not try the algorithm for periods affected by the row anomaly. Regarding how often to train, the algorithm would need to be retrained whenever there are instrumental changes, for example, drifting wavelength-dependent calibration or increased noise or other artifacts. This would need to be assessed on a case-by-case basis. We added a paragraph in the section on practical implementation issues:**

**"Another consideration is how often a NN would need to be retrained. If the instrument were spectrally stable, retraining might not be necessary or might be infrequent. However, destriping may still be necessary to correct for transient spectral artifacts. Retraining should be done whenever there is a substantial change in the instrument spectral characteristics. Since the OMNO2 algorithm uses monthly-averaged solar irradiances, it may be more optimal to similarly normalize with respect to the same set of solar irradiances before training than to use only the radiances as we have done here as the solar data may help to account for instrumental changes."**

*p19, line 392: it would be good to add these results in SI.

**We added a table showing the statistics in the main part of the paper.**

Typos

-P1, line4: S5P resolution is 3.5 km x 5.5 km (not 3.5 km x 5 km)

**Fixed, thank you.**

-p4, line94: 'gloyoxal'-> 'glyoxal'

**Fixed, thank you.**

-p4, line94: 'spectral imprint' -> unclear what is meant by 'imprint'

**We changed "imprint" to "signature". We hope that is clearer.**

---

## Author Response (AR2)

Dear editor:

The only changes made to the manuscript were to add parenthesis around subfigures (e.g., Figure 2a changed to Figure 2(a) and to correct one typo in the Figure 5 caption: change "2)" to "(a)".

Sincerely,
Joanna Joiner